# Mineral Formation during Shipboard Ocean Alkalinity Enhancement Experiments in the North Atlantic

3 4

1 2

Mohammed S. Hashim<sup>1</sup>, Lukas Marx<sup>1</sup>, Frieder Klein<sup>1</sup>, Chloe L. Dean<sup>1,2</sup>, Emily Burdige<sup>1,2</sup>, Matthew Hayden<sup>1</sup>, Daniel C. McCorkle<sup>3</sup>, and Adam V. Subhas<sup>1</sup>

9

101112

Corresponding author: Mohammed S. Hashim; Email: mohammed.hashim@whoi.edu

1314

15

# Keywords

marine carbon dioxide removal, ocean alkalinity enhancement, mineral precipitation, climate solutions, aragonite, brucite,

16 17

#### **Abstract**

Ocean alkalinity enhancement (OAE) is a carbon dioxide (CO<sub>2</sub>) removal approach that involves the addition of alkaline substances to the marine environment to increase seawater buffering capacity and allow it to absorb more atmospheric CO<sub>2</sub>. Increasing seawater alkalinity leads to an increase in the saturation state  $(\Omega)$  with respect to several minerals. This may trigger mineral precipitation, consuming the added alkalinity and decreasing the overall efficiency of OAE. To explore mineral formation due to alkalinity addition, we present results from shipboard experiments in which an aqueous solution of NaOH was added to unfiltered seawater collected from the surface ocean in the Sargasso Sea. Alkalinity addition ranged from 500 to 2000 µmol.kg<sup>-1</sup> <sup>1</sup> and the carbonate chemistry was monitored through time by measuring total alkalinity (TA) and dissolved inorganic carbon (DIC), which were used to calculate  $\Omega$ . The amount of precipitate and its minerology were determined throughout the experiments. Mineral precipitation took place in all experiments over a timescale of hours to days. The dominant precipitate phase is aragonite with trace amounts of calcite and magnesium hydroxide (MgOH<sub>2</sub>, i.e., brucite). Aragonite crystallite size increases and its micro-strain decreases with time, consistent with Ostwald ripening. The precipitation rate (r) in our experiments and those of other calcium carbonate (CaCO<sub>3</sub>) precipitation OAE studies correlates with aragonite saturation state ( $\Omega_A$ ), and the resulting fit of  $\log_{10}(r) = n \times \log_{10}(\Omega_A - 1) + \log(k)$  yields a reaction order  $n = 2.15 \pm 0.50$  and a rate constant k = $0.20 \pm 0.10 \,\mu\text{mol.hr}^{-1}$ . The reaction order is comparable to that derived from previous studies, but

<sup>&</sup>lt;sup>1</sup> Department of Marine Chemistry and Geochemistry, Woods Hole Oceanographic Institution, Woods Hole, MA, USA

<sup>&</sup>lt;sup>2</sup> MIT-WHOI Joint Program in Oceanography/Applied Ocean Science & Engineering, Cambridge and Woods Hole, MA, USA

<sup>&</sup>lt;sup>3</sup> Department of Geology and Geophysics, Woods Hole Oceanographic Institution, Woods Hole, MA, USA

the rate constant is an order of magnitude lower, which we attribute to the fact that our experiments are unseeded compared with previous studies that used aragonite seeds which act as nuclei for precipitation. Observable precipitation was delayed by an induction period, the length of which is inversely correlated with the initial  $\Omega$ . Mineral precipitation occurred in a runaway manner, decreasing TA to values below that of seawater prior to alkalinity addition.

This study demonstrates that the highest risk of mineral precipitation is immediately following alkalinity addition and before dilution and  $CO_2$  uptake by seawater, both of which lowers  $\Omega$ . Aragonite precipitation will decrease OAE efficiency, because aragonite is typically supersaturated in surface ocean waters. Thus, once formed, aragonite essentially permanently removes the precipitated alkalinity from the  $CO_2$  uptake process. Runaway mineral precipitation also means that mineral precipitation following OAE may not only decrease OAE efficiency at sequestering  $CO_2$  but could render this approach counterproductive. As such, mineral precipitation should be avoided by keeping  $\Omega$  below the threshold of precipitation and quantifying its consequences on OAE efficiency if it occurs. Lastly, in order to be able to quantitatively determine the impact of mineral precipitation during OAE, a mechanistic understanding of precipitation in the context of OAE must be developed.

# 54 1 Introduction

The concentration of atmospheric carbon dioxide ( $CO_2$ ) continues to increase due to human activity, leading to an increase in global mean temperature by > 1 °C since pre-industrial times (Cannon, 2025; Hawkins et al., 2017). Nearly 30% of anthropogenic  $CO_2$  emitted annually into the atmosphere is absorbed by the oceans (Friedlingstein et al., 2022; Gruber et al., 2019), decreasing seawater pH, thus causing ocean acidification (Doney et al., 2009). Rapid reductions in anthropogenic  $CO_2$  emissions are necessary to minimize the negative impacts of global warming and ocean acidification. Such emissions reductions will need to be supplemented by active removal of atmospheric  $CO_2$  in order to meet climate objectives (e.g., Paris Climate Agreement, 2015) and offset  $CO_2$  emissions from difficult-to-abate sectors. Ocean alkalinity enhancement (OAE) is a marine-based  $CO_2$  removal (mCDR) approach that involves the addition of alkaline substances to the surface ocean or to surficial sediments to increase seawater buffering capacity, allowing it to absorb more atmospheric  $CO_2$  (Renforth and Henderson, 2017).

Adding alkalinity to seawater lowers its partial pressure of  $CO_2$ . As alkalized seawater equilibrates with the atmosphere, it takes up additional atmospheric  $CO_2$ , which increases seawater dissolved inorganic carbon (DIC) and brings about a new steady state. Alkalinity could be enhanced by adding an alkaline solution or minerals that release alkalinity upon dissolution in seawater (Eisaman et al., 2023). Increasing seawater alkalinity drives an increase in the saturation state ( $\Omega$ ) with respect to several minerals (Fig. 1; Hartmann et al., 2023; Moras et al., 2022; Schulz et al., 2023), which is given by:

$$\Omega = \frac{IAP}{k_{sp}} \tag{1}$$

where IAP is the ionic activity product and  $K_{sp}$  is the mineral solubility product. An  $\Omega > 1$  indicates supersaturation,  $\Omega < 1$  indicates undersaturation, and  $\Omega = 1$  indicates chemical equilibrium between the solid and solution. For calcium carbonate minerals (vaterite, aragonite, calcite) and amorphous calcium carbonate, the increase in  $\Omega$  occurs because adding alkalinity to seawater raises its pH and shifts the carbonate chemistry speciation towards higher concentration of carbonate ions ([CO<sub>3</sub><sup>2</sup>-]). For metal hydroxide minerals, such as magnesium hydroxide (i.e., brucite), the increase in  $\Omega$  is due to the increase in the concentration of [OH-]. An increase in  $\Omega$  due to alkalinity addition can trigger mineral precipitation, which consumes some or all of the added alkalinity, thus decreasing the efficiency of OAE at sequestrating atmospheric CO<sub>2</sub>. Therefore, understanding the timing and kinetics of mineral formation in the context of OAE will help devise an implementation plan that maximizes CO<sub>2</sub> sequestration.

Fig. 1. The saturation state ( $\Omega$ ) with respect to a number of selected minerals calculated using Geochemist's Workbench software for a solution of average seawater composition at 27 °C and 1 bar. The horizontal dashed line represents equilibrium, above which the solution is supersaturated and below which the solution is undersaturated. The triangles at the top of the plot denote the conditions for each experiment. The calculations assumed a closed system where seawater remained unequilibrated with atmospheric CO<sub>2</sub>.

Precipitation is thermodynamically favored if the solution is supersaturated ( $\Omega > 1$ ). However, minerals do not always precipitate once  $\Omega$  exceeds 1. Typically, there exists a higher, mineral-specific  $\Omega$  threshold above which precipitation occurs (Pokrovsky, 1998). This threshold is higher for homogenous (i.e., spontaneous) precipitation, where nuclei form directly from the solution, than for heterogeneous precipitation, where minerals nucleate onto existing surfaces (Morse et al., 2007). The exact  $\Omega$  threshold and the factors influencing it are not fully understood. Determining the  $\Omega$  threshold is particularly challenging for OAE-induced precipitation, given that alkalinity might be added in the form of minerals that would provide nucleation sites onto which secondary precipitation can take place, potentially lowering the threshold  $\Omega$  for precipitation.

The kinetics of precipitation are also important. Precipitation does not necessarily occur as soon as  $\Omega$  is above the precipitation threshold, but may be delayed by a period of time referred to as the induction period (Hashim et al., 2023; Kaczmarek et al., 2017; Pokrovsky, 1998). For OAE, if the induction period is longer than the time it takes seawater to equilibrate with atmospheric  $CO_2$  (e.g., Jones et al., 2014) or the time of alkalinity dilution (He and Tyka, 2023), then the risk of secondary precipitation is minimized. Hence, the induction period is a critical parameter to incorporate into models of alkalinity consumption and will be an important factor when balancing pH thresholds, dilution timescales, and  $CO_2$  uptake rates. Thus far, little attention has been given to characterizing induction periods, particularly in OAE relevant studies.

The precipitation of each mineral has a unique impact on seawater carbonate chemistry. For example, the precipitation of 1 mole of brucite (Mg(OH)<sub>2</sub>) decreases alkalinity by 2 moles and does not impact DIC, whereas the precipitation of 1 mole of aragonite (CaCO<sub>3</sub>) decreases alkalinity by 2 moles and DIC by 1 mole. The type of mineral that precipitates is not only dictated by  $\Omega$  but also by a range of thermodynamic and kinetic factors including temperature, and the presence and concentration of chemical catalysts and inhibitors (Burton and Walter, 1987; Hashim and Kaczmarek, 2021; Morse et al., 1997; Morse and He, 1993; Subhas et al., 2017). The compounding effects of these variables lead to the phenomenon — often explained qualitatively by the Ostwald Step Rule (Morse and Casey, 1988) — that what precipitates from seawater is not necessarily the most thermodynamically stable phase, but the most kinetically favored. For carbonates, this is exemplified by the observation that while dolomite and calcite are the most stable phases, aragonite and high-Mg calcite are the ones that are more likely to precipitate. The impact of alkalinity addition on the mineralogical precipitation landscape remains unclear. It is also unknown how

minerals change through time after alkalinity is added to seawater and whether they will sink and export the alkalinity to the deep ocean, thus decreasing OAE efficiency, or redissolve over time, releasing the alkalinity back into the surface ocean for CO<sub>2</sub> sequestration.

Here we present results from experiments whereby alkalinity was added as an aqueous solution of NaOH to seawater collected from the surface ocean and incubated on deck in a flowing-seawater temperature bath. The carbonate chemistry was monitored through time by measuring TA and DIC, which were used to calculate  $\Omega$ . The mineralogy of the precipitate was determined using X-ray diffraction.

#### 2 Methods

### 2.1 Experimental setup

Experiments were conducted during a research expedition aboard the R/V Atlantic Explorer in the Sargasso Sea (31°40'00"N, 64°10'00"W) near the Bermuda Atlantic Time Series site (BATS) from September 5<sup>th</sup> to 11<sup>th</sup>, 2023. The experiments were performed in opaque 5L Cali-5-Bond<sup>TM</sup> multi-layer foil bags (Calibrated Instruments), placed in a flow-through incubator where surface seawater was continuously flowed to maintain a constant temperature of approximately 27 °C. Each bag was filled with approximately  $3 \pm 0.02$  L of unfiltered seawater using a rubber hose that was flushed with water to remove air bubbles, ensuring that no air entered the bags. Bags were rinsed 3 times with seawater before filling and sealing with the Luer-fitted stopcock. The bags were allowed to float freely in the incubator, and they moved continuously due to the ship movement. The mixed layer at the study site was approximately 40 m thick, and seawater for the experiment was collected from the upper 10 m. The seawater was not filtered nor was it passed through a mesh in order to mimic a realistic OAE scenario. The total particle concentration in the seawater was 0.15 mg/L and the particulate inorganic carbon (PIC) concentration was 0.13  $\mu$ mol/L.

Experiments involved the addition of NaOH solution prepared by weighing ACS grade NaOH in the lab prior to the cruise in a plastic Falcon tube that was capped and sealed with parafilm tape. During the cruise, DI water was added to make up stock NaOH solutions with a final concentration of 1 M. The NaOH solution was pipetted into the seawater filled bags through the Luer-fitted stopcock. Because NaOH contributes only alkalinity but not DIC, seawater in the experiments was out of equilibrium with the atmosphere, which was intended to simulate conditions immediately following alkalinity addition to seawater during OAE deployments.

In total, 5 experiments were conducted (Table 1). The first experiment (experiment A in Table 1) was a control with no alkalinity addition. In the second, third and fourth experiment (B, C, and D) alkalinity was enhanced by 500, 1000, and 2000 µmol.kg<sup>-1</sup> respectively. The fifth experiment (E) represents a set of "sacrificial" time series experiments whereby 9 bags were prepared similar to other experiments and alkalinity was enhanced by 1000 µmol.kg<sup>-1</sup> in each one of them, but each bag was sequentially opened and filtered in order to evaluate the precipitate mineralogy through time. In experiment E, water samples for TA and DIC measurements were taken only at the end of the experiment. The experiments were run for approximately 5 days.

Table 1. Experimental setup and conditions<sup>†</sup>.

| Experiment | Alkalinity<br>addition (μmol.kg <sup>-</sup> 1) | рН   | Saturation state with aragonite ( $\Omega_A$ ) | Duration (h)      | Experiment description in legends |
|------------|-------------------------------------------------|------|------------------------------------------------|-------------------|-----------------------------------|
| A          | 0                                               | 8.25 | 5.3                                            | 122.9             | (A) Control                       |
| В          | 500                                             | 8.68 | 11.3                                           | 125.5             | (B) +500                          |
| С          | 1000                                            | 9.02 | 17.5                                           | 123.1             | (C) +1000                         |
| D          | 2000                                            | 9.71 | 28.0                                           | 123.9             | (D) +2000                         |
| E*         | 1000                                            | 9.02 | 17.5                                           | Various durations | (E) +1000                         |

<sup>†</sup> The seawater used for the experiments has an average TA of  $2547 \pm 10$ , DIC of  $2082 \pm 26 \,\mu\text{mol.kg}^{-1}$ , and salinity of 36 psu. The experimental temperature was 27 (°C).

#### 2.2 Sampling from experiments

Two separate 12 mL seawater samples were taken from bags through time, one for DIC and one for TA. Each of these samples were subsequently modified in order to test recently proposed best practices for carbonate chemistry sampling techniques (Schulz et al., 2023). These proposed techniques were designed to retain the original DIC and TA values at the time of sampling while decreasing  $\Omega$  in the sample container to avoid mineral precipitation during sample storage. For DIC samples, adding an acid to a sample in a completely sealed vessel with no headspace neutralizes a proportion of the previously added alkalinity and thus decreases  $\Omega$  while retaining all the DIC inside the vial. Similarly for TA samples, bubbling  $CO_2$  into the sample increases the DIC, and thus decreases  $\Omega$  without changing the TA. As such,  $\Omega$  can be lowered in both samples to prevent mineral precipitation during sample storage in a way that allows for the accurate determination of DIC and TA (Schulz et al., 2023). We note that these techniques only work for conservative carbonate system parameters (i.e. DIC and TA), and not for non-conservative parameters such as  $pCO_2$  or pH.

<sup>\*</sup> This experiment represents a time series whereby 9 experiments (bags) were started at the same time, and they were opened and filtered sequentially at specific time points to determine precipitate mineralogy.

The 12 mL aliquot taken for DIC was passed through a 0.2 µm filter into a gas-tight borosilicate vial (CHROMONE®, NJ, USA), poisoned with 2.4 µL of saturated HgCl<sub>2</sub>, and then acidified by adding a pre-calculated volume of 0.075 M HCl using a glass syringe through the plastic vial septum to titrate the initially added alkalinity. Seawater and acid injections in the vial were done steadily to minimize gas exchange. The amount of HCl added was 80, 160, 400, and 160 µl for samples taken from experiments B, C, D, and E, respectively. The 12 mL TA aliquot was filtered (0.2 µm filter), and then bubbled with pure CO<sub>2</sub> using a nylon tubing with a stainless steel needle for 30 seconds to increase its DIC without changing TA, followed by poisoning with HgCl<sub>2</sub>. A gas regulator was used to maintain a constant CO<sub>2</sub> flow rate and to prevent over-bubbling. The DIC and TA samples were returned to the lab where they were kept in cool and dark conditions until analysis, which took place within 2 months.

Experiments were quenched by filtering all remaining seawater through 47 mm 0.8  $\mu$ m polycarbonate filters using a peristaltic pump. Filters were rinsed with deionized and purified water (18.2 M $\Omega$ ), dried at 55°C, and stored cool and in the dark. The precipitates were then scraped off the filters and analyzed for mineralogy with X-ray diffraction (see Section 2.4).

# 2.3 TA and DIC measurements and saturation state calculations

TA was determined using an open-system Gran titration on weighed 5 mL samples in duplicate using a Metrohm 805 Dosimat, with a 1 mL burette, and an 855 robotic Titrosampler. An 0.04 M HCl titrant was used to first acidify the sample to a pH of 3.9 before continuing to a pH of 3.25, dosing at 0.02 mL increments. The analyses were calibrated using in-house seawater standards that were ran every 15 samples, to assess titrant and electrode drifts throughout the day. A nonlinear least-squares method was used to determine TA as outlined in the Best Practices guide (Dickson et al., 2007).

DIC was determined using an Apollo LI-5300A connected to a Li-COR CO<sub>2</sub> analyzer, with CO<sub>2</sub> extracted from a 1.5 mL sample volume by adding 0.8 mL of 3% phosphoric acid. Once opened, the sample lines were inserted to the base of the vial and sealed with parafilm tape to limit gas exchange. Before each analysis, 0.75 mL of sample and 0.8 mL of acid is drawn into the sample syringe to flush out any prior remnants from the system. After the flush, the 1.5 mL sample is drawn into the calibrated syringe and injected into the reaction chamber, where resulting CO<sub>2</sub> is carried by a zero CO<sub>2</sub> air stream to the Li-COR CO<sub>2</sub> analyzer. Samples were run in triplicates. The

instrument was calibrated twice daily against an in-house seawater standard that were intercalibrated against seawater Certified Reference Materials (Dickson batch #187).

The saturation state with respect to aragonite ( $\Omega_A$ ) throughout the experiments was calculated using PyCO<sub>2</sub> 1.8.1 (Humphreys et al., 2022), the Python version of the original CO2SYS program (Lewis et al., 1998) using the carbonic acid dissociation constants of Mehrbach et al. (1973) refitted by Dickson and Millero (1987). The  $\Omega_A$  calculations used a corrected concentration of Ca to account for changes induced by CaCO<sub>3</sub> precipitation, calculated according to the following equation:

$$[Ca] = [Ca]_{s\_corr} + (\frac{TA_{msrd} - TA_{initial}}{2})$$
 (2)

Where  $[Ca]_{s\_corr}$  is the mean seawater concentration of Ca in mmol.kg<sup>-1</sup> corrected for salinity by multiplying 10.28 by measured salinity at BATS and dividing by 35,  $TA_{msrd}$  is the measured TA in mmol.kg<sup>-1</sup> in each sample, and  $TA_{sw}$  is the initial TA after alkalinity addition in mmol.kg<sup>-1</sup>. The saturation state with respect to other minerals was calculated using the Geochemist's Workbench (GWB) software, version 11 using the thermo.dat database at 1 bar and 27 °C (Bethke, 1996) for seawater, equivalent to that used in our experiments, titrated with 1 M NaOH solution under closed system conditions (Fig. 1). GWB was used to calculate  $\Omega$  with respect to a wider range of minerals than possible using PyCO<sub>2</sub>. The carbonate chemistry data are archived in Hashim et al. (2025a).

### 2.4 Mineralogy

The solid precipitates collected by filtering the seawater from the experimental bags were analyzed for mineralogy using X-ray diffraction (XRD). Precipitates were gently powdered by hand using an agate mortar and pestle under acetone and mounted on a silicon zero-background diffraction plate, which was placed in an automatic sample changer (Hashim and Kaczmarek, 2022). Samples were measured at the MIT.nano facility at the Massachusetts Institute of Technology using a PANalytical X'Pert PRO X-ray powder diffractometer using a Cu anode and an X'Celerator Scientific 1D position-sensitive detector in Bragg-Brentano geometry. The 2θ range was 5 to 100° and a count time of 1 s.step-1. Data were processed using the fundamental-parameters Rietveld refinement program TOPAS V7 (Coelho, 2018). The data were corrected for an instrument zero error determined using the NIST standard LaB<sub>6</sub> (660c), which has a certified unit-cell parameter of 4.156826 Å, crystallite size (Lvol) of 500 nm, and no micro-strain related peak broadening (Black et al., 2020). The zero error refers to a shift in diffraction patterns due to

misalignment of the detector. Instrument parameters were set to known values during Rietveld refinement and TOPAS was used to determine relative mineral abundances, unit cell parameters, crystallite size, and micro-strain (Bish and Howard, 1988). All raw XRD data and the Rietveld refinement results are archived in Hashim et al. (2025b).

A TA Instruments Q600 simultaneous thermal analyzer was used for thermogravimetric analysis and differential scanning calorimetry of mineral precipitates from Experiment D. About 35 mg was heated from room temperature to  $1100~^{\circ}$ C with a heating rate of  $10~^{\circ}$ C per minute in a  $N_2$  atmosphere. The  $N_2$  flow rate was 50 mL.min<sup>-1</sup>. Routine measurements of standard materials supplied by the manufacturer indicate that the temperature is accurate to  $1~^{\circ}$ C and the weight change to  $0.5~\mu g$ . The raw TGA data are archived in Hashim et al. (2025c).

## 3 Results

# 3.1 Precipitate mineralogy

Relative mineral abundances derived from X-ray diffraction (XRD) data suggest the precipitation of aragonite, calcite, and halite (Fig. 2A). Halite presence can be attributed to its precipitation as a result of seawater evaporation during filter drying and is omitted from relative abundance comparisons (Fig. 2B). The relative abundance of aragonite and calcite varies slightly across experiments in that calcite abundance decreases with the increase of alkalinity addition (Fig. 2B). In experiment E, aragonite remains the dominant phase through time (Fig. 3). Furthermore, results from the Rietveld refinement for experiment E (Table 1) samples reveal that aragonite crystallite size increases whereas the lattice strain decreases with time (Fig. 4). Precipitates at the end of experiments C and D (+1000 and 2000 µmol.kg<sup>-1</sup>) yielded aragonite with smaller crystallite size that falls of the trend of experiment E (Fig. 4A).

**Fig. 2. A)** Raw X-ray diffraction (XRD) spectra of precipitate from experiment B, C, and D truncated between 25 and 35° 2theta for easier viewing. All three precipitates were collected at the end of the experiments which were conducted for ~ 123 h. Miller indices of prominent peaks are shown where A stands for aragonite, H for halite, and C for calcite. **B)** Relative mineral abundance determined via Rietveld refinement of the three spectra shown in (A). Each of aragonite and calcite abundances are normalized to the total (aragonite + calcite) given that halite is considered an artifact that precipitated during filter drying.

**Fig. 3.** Raw XRD data collected for samples from experiment E where alkalinity was enhanced by 1000 μmol.kg<sup>-1</sup>. Each scan was collected for a sample filtered at a certain period of time from the onset of the experiment. Miller indices of prominent peaks are shown where A stands for aragonite and H for halite. Raw XRD data is available in Hashim et al. (2025b).

Thermogravimetric analysis of precipitate (Fig. 5) from experiment D revealed two weight loss events that can be attributed to dehydroxylation of magnesium hydroxide (2.97 wt.% at ~353 °C) and calcination of Ca-carbonate (33.52 wt% at ~768 °C) (Földvári, 2011; Klein et al., 2020). Additional weight loss occurred at 467 °C (0.98 wt.%) and 968 °C (3.88 wt.%); the underlying cause of these weight loss events remains unresolved but we speculate that the minor weight change at 467 °C could be attributed to the loss of adsorbed water during the phase transition of aragonite to calcite, or possibly to decomposition of a hydrous Mg-carbonate mineral similar to hydromagnesite.

**Fig. 4.** Crystallographic characteristics of aragonite precipitate determined via Rietveld refinement. **A)** Aragonite crystallite size plotted against time. XRD data were collected for precipitate in experiment E thought time and only at the end in experiments C and D. The logarithmic function is fitted through the E data only. The *p-value* for the fit is  $\ll 0.05$  and  $R^2$  is 0.76. **B)** Aragonite micro-strain as a function of time for the same experiments shown in (A). The logarithmic function is fitted through experiment E data only. The *p-value* of the fit is 0.02 and  $R^2$  is 0.47. The shaded region represents the 95% confidence interval.

### Carbonate chemistry response to alkalinity enhancement

The measured TA and DIC of the control experiment with no alkalinity addition remain nearly constant throughout the duration of the experiment with an average value ( $\pm$  1 $\sigma$ ) for TA of 2547  $\pm$  10 µmol.kg<sup>-1</sup> and DIC of 2082  $\pm$  26 µmol.kg<sup>-1</sup> (Fig. 6A and 6B). All alkalinity enhancement experiments show elevated initial TA values as expected (Fig. 6A) and exhibit a gradual decrease in TA, DIC, and  $\Omega_A$  with time (Figs. 6A, 6B, and 7A). The decrease in TA, DIC, and  $\Omega_A$  is steeper and reaches lower final values for experiments with higher alkalinity addition. The lowest final TA value is observed for experiment D with highest alkalinity addition of 2000 µmol.kg<sup>-1</sup>, followed by experiment C (1000 µmol.kg<sup>-1</sup>), followed by experiment B (500 µmol.kg<sup>-1</sup>) (Fig. 6A). Similar trends are observed with DIC, which decreases more steeply and reaches the lowest final values for experiments where alkalinity is enhanced the most (Fig. 6B).

Fig. 5. Thermogravimetric analysis of precipitates from experiment D ( $\pm$ 2000  $\mu$ mol.kg<sup>-1</sup>) after 124 h. The data is consistent with the presence of Ca-carbonate (likely aragonite) and minor amounts of brucite, which are indicated by the shaded area. The small peak around 470°C denoted with a question mark is suggestive of the presence of hydromagnesite which was initially supersaturated. Alternatively, this event could be due to loss of minor adsorbed water during the transition of aragonite to calcite. The raw TGA data is available in Hashim et al. (2025c).

TA and DIC decrease linearly with each other with slopes of 1.39, 1.44, and 1.85 for experiment B, C, and D, respectively (Fig. 6C). Such slopes are difficult to interpret in the context of mineral precipitation given that  $CaCO_3$  minerals (aragonite, calcite, vaterite) should decrease TA and DIC in a 2:1 ratio (slope = 2) whereas magnesium hydroxide (i.e., brucite) decreases only TA (slope =  $\infty$ ). The slope values being < 2 imply that more DIC is removed than can be explained by  $CaCO_3$  precipitation (Fig. 6D). We suggest an explanation for these observations in Section 4.4.

#### 3.3 Amount and rate of mineral precipitation

Measured TA through time was used to estimate the amount of CaCO<sub>3</sub> precipitate (Fig. 8) assuming that the decrease in TA with time was solely due to CaCO<sub>3</sub> precipitation and a TA to CaCO<sub>3</sub> molar ratio of 2:1, according to the following equation:

$$m_{CaCO_3} = \frac{TA_{initial} - TA_{msrd}}{2} \times M_{CaCO_3}$$
 (3)

Where  $m_{CaCO_3}$  is the mass of CaCO<sub>3</sub> precipitate in  $\mu$ g.kg<sup>-1</sup> and  $M_{CaCO_3}$  is the molar mass of CaCO<sub>3</sub> in g.mol<sup>-1</sup>. Assuming that TA decreases only due to CaCO<sub>3</sub> precipitation is reasonable

because the dominant mineralogy in all experiments is aragonite (see Section 3.1). These calculations indicate that a higher amount of precipitation occurs in experiments where TA was enhanced the most (Fig. 8). Moreover, in experiment B (+500 µmol.kg<sup>-1</sup>), precipitation occurs gradually throughout the duration of the experiment whereas in experiments C and D (+1000 and 2000 µmol.kg<sup>-1</sup>), most precipitation takes place during the first 24 hours followed by a gradual increase in the amount of precipitate with time (Fig. 8).

The rate of mineral precipitation was calculated using the following equation:

$$r = \frac{\Delta T A}{2 \times \Delta t} \tag{4}$$

Where r is the rate in  $\mu$ mol.h<sup>-1</sup>,  $\Delta$ TA is the change in TA between two datapoints, and  $\Delta$ t is the time difference between the same datapoints (Fig. 9). The rate data were fitted to an empirical rate law in its logarithmic form (Zhong and Mucci, 1989):

$$log(r) = n \times log(\Omega_a - 1) + log(k)$$
 (5)

Where n is the empirical reaction order and k is the rate constant in  $\mu$ mol.hr<sup>-1</sup>. Plotting log (r) against log ( $\Omega$  –1) shows that there is a relationship for our experiments, experiments of Moras et al. (2022), and aragonite precipitation experiments of Mucci et al. (1989). A fit through our and Moras et al. (2022) data yields a reaction order of 2.15  $\pm$  0.40 and a rate constant of 0.20  $\pm$  0.10  $\mu$ mol.hr<sup>-1</sup>. A fit through Mucci et al. (1989) data gives a reaction order of 1.5  $\pm$  0.3 and a rate constant of 45  $\pm$  2  $\mu$ mol.hr<sup>-1</sup>.

Fig. 6. A) Measured TA through time showing a gradual decrease from the initial enhanced value except for the control experiment. B) Measured DIC (solid line) and calculated DIC (dashed line) using Eq. 6. Both measured and calculated DIC decrease with time from the initial seawater value except for the control experiment. C) TA plotted against measured DIC with linear regressions fitted for experiment B, C, and D (equations shown in the upper left corner). The grey lines represent the expected slopes as a result of  $CaCO_3$  (slope = 2) and brucite (slope =  $\infty$ ) precipitation. D) Measured DIC plotted against DIC calculated assuming 2:1 TA:DIC removal from  $CaCO_3$  precipitation, showing that most datapoints have lower measured than calculated DIC and hence deviate from the 1:1 line suggesting that some DIC was removed during HCl addition which was done to preserve the samples.

Fig. 7. The saturation state with respect to aragonite ( $\Omega_A$ ) for experiments from this study calculated using measured TA and calculated DIC (A) and from Moras et al. (2022) (B). The lines represent a model fitted using Eq. 8. Both data and model are characterized by an induction period ( $t_{ip}$ ) where  $\Omega_A$  remains similar to the initial value, followed by an exponential decrease, followed by an asymptote value where  $\Omega_A$  ceases to change with time (extrapolated at infinite time  $\Omega_{\infty}$ ). Note the difference in the range of values of x and y axes between A and B.

Fig. 8. The amount of CaCO<sub>3</sub> precipitate through time estimated based on changes in TA using eq. 3.

#### 4 Discussion

# 4.1 Thermodynamic constraints on mineral precipitation

The  $\Omega$  of the seawater used in our experiments both prior and following alkalinity addition was calculated with respect to a set of minerals that includes aragonite (CaCO<sub>3</sub>), calcite (CaCO<sub>3</sub>),  $(CaMg(CO_3)_2),$ brucite (Mg(OH)<sub>2</sub>), magnesite (MgCO<sub>3</sub>), hydromagnesite dolomite (Mg<sub>5</sub>(CO<sub>3</sub>)<sub>4</sub>(OH)<sub>2</sub>·4H<sub>2</sub>O), and huntite (Mg<sub>3</sub>Ca(CO<sub>3</sub>)<sub>4</sub>) (Fig. 1). This list represents only a small fraction of all the minerals that could possibly precipitate but this set of minerals was chosen because they have been reported to occur naturally in marine environments and some of them (e.g., aragonite and brucite) have been shown to precipitate during OAE experiments (Bach et al., 2024; Hartmann et al., 2023; Moras et al., 2022). Seawater was initially highly (i.e.,  $\Omega > 100$ ) supersaturated with respect to dolomite and huntite and moderately (i.e.,  $\Omega \sim 10$ ) supersaturated with aragonite, calcite, and magnesite (Fig. 1). In contrast, brucite and hydromagnesite are undersaturated in natural seawater, and become supersaturated as alkalinity increases. Alkalinity addition increases  $\Omega$  with respect to all of these minerals. Hydromagnesite becomes supersaturated when the TA reaches ~ 2700 µmol.kg<sup>-1</sup> and brucite becomes supersaturated when the TA reaches  $\sim 3100$  and becomes more supersaturated than aragonite at TA of  $\sim 4600$  µmol.kg<sup>-1</sup> (Fig. 1). The reason for the different saturation behaviors is that magnesium hydroxide solubility is a function of [OH-]2, whereas CaCO3 mineral solubilities are proportional to [CO32-]. Hydroxide ion concentration increases with TA addition, thus rapidly increasing the saturation state of hydroxidebearing minerals.

# 4.2 Mineralogy of the OAE induced precipitation

In experiments B, C, and D (+500, +1000, +2000 µmol.kg<sup>-1</sup>, respectively), the dominant precipitate mineralogy according to XRD data is aragonite, followed by a small amount of calcite (Fig. 2). This observation is consistent with previous studies showing that the calcium carbonate precipitating from seawater at a temperature of 27 °C is aragonite (Burton and Walter, 1987; Hartmann et al., 2023; Hashim et al., 2024; Moras et al., 2022; Morse et al., 1997; Morse and He, 1993). This is often attributed to the inhibition of calcite growth by Mg (Berner, 1975; Mills et al., 2022) that leads to the precipitation of the metastable aragonite at the expense of the more stable calcite, despite the fact that the solution is more supersaturated with respect to calcite than aragonite (Fig. 1).

Aragonite was also the dominant precipitated phase in experiment E (+1000 μmol.kg<sup>-1</sup>) where mineralogy was determined through time (Fig. 3). The crystallographic characteristics of the aragonite evolved throughout the experiment, specifically, aragonite crystallite size increased whereas its micro-strain decreased (Fig. 4). Crystallite size refers to a coherently diffracting domain within a crystal and micro-strain refers to distortions or deformations within the crystal lattice arising from dislocations, point defects, or crystal boundaries (Bish and Post, 2018; Hashim et al., 2023; Mittemeijer and Welzel, 2008). These data are consistent with the process of Oswald ripening whereby aragonite recrystallizes to a more stable phase with a larger crystallite size and lower strain. It is unclear why experiments C and D yielded aragonite with smaller crystallite size after a longer period of time compared to experiment E (Fig. 4A). The higher alkalinity addition in experiment D compared to E may suggest that there is an inverse relationship between alkalinity (or  $\Omega$ ) and aragonite crystallite size. However, this cannot explain why experiment C, which used similar amount of alkalinity as experiment E, produced aragonite with smaller crystallite size. Future work should explore the crystallographic properties of precipitated minerals under a wide range of alkalinity additions. We also suggest supporting the XRD-derived crystallite size measurements with other approaches such as Transmission Electron Microscopy. Nonetheless, the changing crystallographic properties of aragonite through time suggests that aragonite recrystallizes to become more stable. In the context of OAE, if aragonite becomes more stable, it will take longer to redissolve and may sink in the water column, decreasing the efficiency of OAE.

Another mineral that could have formed in experiment E is brucite, which was initially slightly supersaturated ( $\Omega_{brucite} \sim 1.7$ ) (Fig. 1A). This slight supersaturation may not have been high enough for brucite to precipitate, perhaps due to kinetic nucleation limitations similar to carbonate minerals. Alternatively, brucite, or more generally amorphous magnesium hydroxide, may have formed initially but redissolved before the first sample was taken at 0.6 h following alkalinity addition. The alkalinity decreased from the initial enhanced value of 3500 to  $\sim 3150 \, \mu \text{mol.kg}^{-1}$  after 0.6 h which corresponds to an  $\Omega_{brucite}$  of 0.7. Thus, brucite could have formed initially but redissolved as aragonite precipitation decreased alkalinity, rendering brucite undersaturated.

In experiment D,  $\Omega_{brucite}$  was initially > 10, i.e., likely sufficient for brucite to precipitate (Fig. 1A). The thermogravimetric analysis (TGA) confirms the presence of a small (~ 3%) amount of brucite in experiment D precipitate (Fig. 5). It is surprising that brucite in experiment D did not redissolve given that at the end of experiment D,  $\Omega_{brucite}$  was < 0.01. It is possible that aragonite

nucleated on brucite crystals, enclosing and protecting them from dissolution. Alternatively, slow dissolution kinetics may have prevented all of the brucite from redissolving, despite near complete undersaturation. It should be noted that brucite was only detected with TGA but not XRD despite that our XRD spectra encompassed the 2θ range of the dominant brucite peaks such as the (001) that typically appears at 18.4° 2θ. It is possible that the small amount of brucite was below the detection limit of XRD, or that the precipitate was amorphous magnesium hydroxide with no diffracting structure rather than brucite mineral. Brucite has been documented to exhibit a wide range of structures including amorphous Mg(OH)<sub>2</sub> with implications for its dissolution kinetics (Cifuentes-Araya et al., 2014; Vermilyea, 1969).

At high  $\Omega$  values, such as those that characterized some of our experiments, vaterite, amorphous calcium carbonate (ACC), and amorphous calcium-magnesium carbonate (ACMC) have been observed to form, but such phases are highly unstable and quickly transform to other more stable phases (Evans et al., 2020; Methley et al., 2024). It is therefore possible that the observed aragonite in our experiment is not the first phase to form but is the product of a transformation reaction from an even less stable phase such as vaterite, ACC or ACMC, although no evidence for the occurrence of these phases was found. A high-resolution sampling within minutes of the start of the experiments as well as using synchrotron XRD could detect the first phases that form at conditions relevant to OAE.

## 4.3 Saturation state $(\Omega)$ calculations and the threshold of precipitation

TA and DIC decrease linearly with each other with a slope of 1.39, 1.44, and 1.85 for experiment B, C, and D (+500, +1000, +2000 µmol.kg<sup>-1</sup>), respectively (Fig. 6C). Slope values < 2 imply that more DIC is removed than can be explained by CaCO<sub>3</sub> precipitation. One way for this to happen is if the acidification of the samples in vials, which was done to lower  $\Omega$  to prevent precipitation during sample storage (Section 2.2), neutralized too much of the sample TA, leading to a high pCO<sub>2</sub> that induced outgassing and thus DIC loss. Accordingly, we calculated DIC using TA by assuming that the decrease in DIC was solely due to the precipitation of CaCO<sub>3</sub> according to the following equation:

$$DIC_i = DIC_o - (\frac{TA_o - TA_i}{2}) \tag{6}$$

Where DIC<sub>o</sub> and TA<sub>o</sub> are the initial DIC and TA values in each experimental series and TA<sub>i</sub> is the value of TA measured throughout the experiment. The assumption that DIC decreases solely due to CaCO<sub>3</sub> precipitation is a reasonable one given that our XRD data show that the dominant

mineralogy in all experiments is aragonite (Fig. 2). In all alkalinity addition experiments, the measured DIC is lower and more variable than the calculated DIC (Fig. 6D). This suggests that in the high alkalinity addition nearly all of the DIC in the sample was lost due to acidification. Thus, the acidification protocol proposed by Schulz et al. (2023) must be carefully evaluated to prevent significant degassing of acidified  $CO_2$  during sample storage, and when the vial is opened for measurement. For these reasons, instead of measured DIC, we use the calculated DIC via equation 6, along with measured TA, to calculate  $\Omega_A$ . For future studies, instead of adding a predetermined amount of acid based on alkalinity addition, we recommend using our and other mineral precipitation studies as a guidance to predict the values of TA and DIC through time, which can then be used to determine the amount of acid to be added such that only excess TA is removed.

The decrease in measured TA, calculated DIC, and  $\Omega_A$  over the course of the experiment (Fig. 6A, 6B, 7A) suggests that precipitation occurs at all alkalinity enhancement levels. This is in general agreement with previous studies showing that precipitation occurs with similar alkalinity additions (Hartmann et al., 2023; Moras et al., 2022). The threshold of saturation state with respect to argonite  $(\Omega_A)$  for homogenous precipitation from seawater is between 13 and 19 (Morse and He, 1993; Pytkowicz, 1973; Sun et al., 2015). For reference, surface ocean seawater has an  $\Omega_A$  of < 5 (Jiang et al., 2015). In our experiments, precipitation is observed at an initial  $\Omega_A$  of 11 where alkalinity was enhanced by 500 µmol.kg<sup>-1</sup>. Since this is the lowest amount of alkalinity added in our study, we suggest that the threshold  $\Omega_A$  for mineral precipitation under the experimental conditions is  $\leq 11$ . This is consistent with previous OAE experiments showing that precipitation takes place at an  $\Omega_A$  of 7 (Moras et al., 2022). Several factors impact the threshold  $\Omega$  for precipitation, one of which is whether precipitation is homogenous or heterogenous (Morse et al., 2007). The threshold  $\Omega$  is higher for homogenous precipitation due to the energy intensive process of forming nuclei out of an aqueous solution (Morse et al., 2007). This is supported by the observation that the threshold  $\Omega$  for precipitation from unfiltered seawater is lower than that from filtered seawater (Hartmann et al., 2023), which means that minerals can nucleate on existing mineral particles in seawater (Alexandersson, 1972).

# **4.4 Precipitation rate**

Using the changes in TA through time (Fig. 6A), we calculated the precipitation rate (r) in our experiments and those of Moras et al. (2022). Plotting this rate against  $\Omega-1$  yielded an empirical reaction order of  $2.15 \pm 0.40$  and a rate constant of  $0.20 \pm 0.10$  µmol.hr<sup>-1</sup> (Fig. 9). The

reaction order is higher but within the uncertainty of that from Mucci et al. (1989) and Burton and Walter (1987) for seeded aragonite precipitation experiments from artificial seawater at 25 °C (Fig. 9), which are  $1.48 \pm 0.3$  and  $1.70 \pm 0.1$ , respectively. The similarity in the reaction order suggests that the precipitation mechanism is similar in our experiments, those of Moras et al. (2022), and the seeded experiments of Mucci et al. (1989) and Burton and Walter (1987).

Fig. 9. The logarithm of precipitation rate (log r) plotted against the logarithm of  $(\Omega - 1)$  for experiments from this study, Moras et al. (2022), and Mucci et al. (1989). The solid black line with the brown shading represents the linear fit through data from our and Moras et al. (2022) experiments with a 95% confidence interval. The equation of this fit provides the reaction order (n = 2.15) and the reaction constant (log k = -0.7; k = 0.2). All rate data are available in Hashim et al. (2025a).

However, the calculated rate constant is lower by about one order of magnitude compared with Mucci et al. (1989). A major difference in the experimental setup is that most previous studies used aragonite seeds to induce precipitation (e.g., Mucci et al., 1989). Aragonite seeds provide a template and thus decrease kinetic limitations to precipitation. In contrast, precipitation in our experiments was homogenous or pseudo-homogenous, where minerals either precipitated directly from seawater, or formed on the inner walls of the foil bags. Nucleating on existing aragonite

crystal is easier than nucleating on plastic or glass material (Subhas et al., 2022). For example, it has been shown that calcite grows faster on existing calcite compared to quartz (Lioliou et al., 2007). In addition, the entire inner surface area of the bag is 0.24 m² whereas the surface area of the aragonite seeds in the experiments of Mucci et al. (1989) is 1.7 m². Normalizing our and Mucci et al.'s rate constants by their respective surface area gives a reaction constant of 0.63 and 26.50 µmol.m⁻².hr for our and Mucci et al (1989) data, respectively. Therefore, the higher rate constant in the experiments of Mucci et al. (1989) are likely due to the presence of aragonite seeds. These comparisons indicate that the availability of nuclei represents a significant control on the rate of precipitation (Morse et al., 2007). They also reveal that the kinetics of homogenous nucleation of calcium carbonate minerals is not fully understood, and that future work should address this knowledge gap, particularly in the context of OAE.

It is worth pointing out that our experiments were conducted at 27 °C, Moras et al. (2022) experiments at 21 °C, and Mucci et al. (1989) at 25 °C. The reaction order is expected to increase by  $\sim 0.066$  per 1 °C and the reaction constant by  $\sim 0.5$  per 1 °C (Burton and Walter, 1987). This suggests that differences in reaction order and constant due to temperature are within the reported uncertainties and are much smaller than the observed differences between our and Moras et al. (2022) experiments and those of Mucci et al. (1989). We have not recalculated all of these results to a single temperature because the study of Burton and Walter (1987), which originally explored the effects of temperature on reaction order and constant, was based on seeded experiments where the reaction constant is much higher than in our and Moras et al. (2022) experiments. This difference in experimental conditions is likely to be a dominant factor, and recalculation to account for slight differences in temperature did not seem warranted. This highlights the need for future work focusing on the role of temperature in mineral precipitation kinetics under conditions representative of a range of OAE scenarios.

#### 4.5 The induction period prior to mineral precipitation

The induction period  $(t_{ip})$  is the time required for precipitation to initiate from a supersaturated solution. In the context of OAE, a longer  $t_{ip}$  allows for more dilution with surrounding seawater during OAE deployments (He and Tyka, 2023), and on longer timescales, allows for more  $CO_2$  uptake to increase DIC (Jones et al., 2014), potentially reducing or even eliminating mineral precipitation. Previous work has shown that the  $t_{ip}$  for  $CaCO_3$  minerals inversely correlates with the initial  $\Omega$  (Pokrovsky, 1998), although the exact mechanism behind

this relationship and how other variables impact  $t_{ip}$  remain unclear. One interesting observation is that higher solution Mg/Ca leads to longer  $t_{ip}$  for the same  $\Omega$  (Fig. 10A), likely due to the inhibition effect of Mg on the nucleation and/or crystal growth of carbonate minerals (Berner, 1975; Bischoff, 1968; Hashim and Kaczmarek, 2020).

Identifying  $t_{ip}$  precisely in highly supersaturated solutions requires a continuous and fast-responding measurements such as pH. To determine the induction period  $(t_{ip})$  as well as the extrapolated  $\Omega$  to infinite time following precipitation, the change in  $\Omega$  over time was empirically described by fitting a Heaviside stepwise function to the data:

$$\Omega = \Omega_o \times (1 - H(t - t_{ip})) + (\Omega_\infty + (\Omega_o - \Omega_\infty) \times e^{-k*(t - t_{ip})} \times H(t - t_{ip}))$$
 (7)

Where  $\Omega$  is the saturation state with respect to aragonite to be calculated through time,  $\Omega_o$  is the initial saturation state in the experiments, which is higher than that of seawater as a result of alkalinity addition, t is time in hours,  $t_{ip}$  is the induction period in hours,  $\Omega_{\infty}$  is the non-zero asymptote value after the exponential decay when  $\Omega$  ceases to change with time, k is a decay constant that is related to the precipitation rate, and H is the Heaviside function which is defined by:

514 
$$H(t) = \begin{cases} 0 & \text{if } t < 0 \\ 1 & \text{if } t \ge 0 \end{cases}$$
 (8)

 When  $t < t_{ip}$  (i.e., during the induction period), the expression  $(t-t_{ip})$  becomes negative, and H(t) equals 0, which makes the first term of the equation equals to  $\Omega_o$ . When  $t > t_{ip}$  (after the induction period), H(t) equals to 1, and  $\Omega$  will decrease with time according to the exponential decay formula in the second term of the equation. This function was fitted to data from experiments B, C, and D (Table 1), as well as to three sets of experiments from Moras et al. (2022) that, similar to our experiments, are characterized by an induction period and exponential decay. The fit was performed by concomitantly changing the free parameters  $\Omega_o, t_{ip}, \Omega_\infty$ , and k while minimizing the difference between the fit and the data using Solver's nonlinear generalized reduced gradient algorithm in Excel®. We note that we only interpret  $t_{ip}$  and  $\Omega_\infty$  in this framework. The exponential part of the equation is a convenient functional form to model the decrease in omega with time, but is purely descriptive of the data and should not be used as a functional relationship between time and the decrease in saturation state. All parameters obtained from the fitting are provided in Table 2.

Table 2. Parameters obtained from fitting Eq. 7 to data for TA and  $\Omega_A$ .

| Parameter                    | Study               | Experiment (TA addition µmol.kg <sup>-1</sup> )        | Initial value | Final (asymptote) | Induction period (h) | k     |
|------------------------------|---------------------|--------------------------------------------------------|---------------|-------------------|----------------------|-------|
| TA (μmol.k g <sup>-1</sup> ) | this study          | B (+500)                                               | 3047.00       | 1912.00           | 10.58                | 0.021 |
|                              | this study          | C (+1000)                                              | 3547.44       | 1486.27           | 7.71                 | 0.181 |
|                              | this study          | D (+2000)                                              | 4547.44       | 1003.41           | 2.44                 | 0.112 |
|                              | Moras et al. (2022) | CaO (+500)                                             | 2721.84       | 1769.90           | 92.10                | 0.009 |
|                              | Moras et al. (2022) | Ca(OH)2 (+500)                                         | 2744.92       | 1608.86           | 159.49               | 0.002 |
|                              | Moras et al. (2022) | Na <sub>2</sub> CO <sub>3</sub> with particles (+1000) | 3394.92       | 2125.22           | 165.50               | 0.009 |
| $\Omega_{\Lambda}$           | this study          | B (+500)                                               | 11.47         | 3.45              | 5.80                 | 0.023 |
|                              | this study          | C (+1000)                                              | 17.73         | 3.13              | 7.44                 | 0.213 |
|                              | this study          | D (+2000)                                              | 28.75         | 2.93              | 2.17                 | 0.131 |
|                              | Moras et al. (2022) | CaO (+500)                                             | 7.52          | 1.83              | 87.92                | 0.010 |
|                              | Moras et al. (2022) | Ca(OH) <sub>2</sub> (+500)                             | 7.32          | 1.00              | 125.74               | 0.002 |
|                              | Moras et al. (2022) | Na <sub>2</sub> CO <sub>3</sub> with particles (+1000) | 9.20          | 1.89              | 117.74               | 0.008 |

Fig. 10. Plots of some of the parameters obtained from fitting equation 7 to data shown in Figure 7. A) Linear-log plot between the induction period  $(t_{ip})$  and the initial  $\Omega$  (all  $\Omega$  values are with respect to aragonite). Circles are data from Pokrovsky (1998) for solutions with different Mg/Ca ratios. B) Alkalinity drawdown ( $\Delta TA = initial\ TA$  following alkalinity addition – final TA at the end of the experiment) plotted against alkalinity addition. The horizontal dashed line is plotted at  $\Delta TA = 0$  indicating no precipitation whereas the dotted line has a slope of 1 indicating that the removed alkalinity due to precipitation equals the added alkalinity. C) Cross plot of the extrapolated  $\Omega$  at infinite time ( $\Omega_{\infty}$ ) and initial  $\Omega$  following alkalinity addition. The 1:1 line indicates that the initial and  $\Omega_{\infty}$  are equal (no precipitation).

For our experiments,  $t_{ip}$  decreases as initial  $\Omega_A$  increases (Fig. 10A). Furthermore, the  $t_{ip}$  of our experiments is shorter than those of Moras et al. (2022) confirming the relationship between  $t_{ip}$  and initial  $\Omega_A$  given that our experiments were conducted at higher  $\Omega_A$  (Fig. 10A). In general, the t<sub>ip</sub> from our experiment B (+500 µmol.kg<sup>-1</sup>) and experiments of Moras et al. (2022) are in good agreement with data from Pokrovsky (1998) for seawater with average Mg/Ca. In contrast, tip of experiments C and D (+1000 and +2000 µmol.kg<sup>-1</sup>) are generally longer than those of Pokrovsky (1998) for the same initial  $\Omega_A$ . Experiment C has a t<sub>ip</sub> of 7 h compared to 1 hr for Pokrovsky's data for the same  $\Omega$  and experiment D has a  $t_{ip}$  of 2 h compared to 0.3 h for Pokrovsky's data (Fig. 10A). These differences could be due to the fact that our tip values are based on a model fitted to the data and do not represent actual measurements. Alternatively, they might be related to the presence of chemical inhibitors. Pokrovsky (1998) used a sulfate- and phosphate-free solution, both of which are known inhibitors of carbonate mineral precipitation and dissolution (e.g., Burton and Walter, 1990; Fernández-Díaz et al., 2010). Given that both our study and Moras et al. (2022) used natural seawater that contains both sulfate and phosphate, we suggest that the inhibition of mineral precipitation by these ions, in addition to Mg, may explain the longer t<sub>ip</sub> in our experiments compared to those of Pokrovsky (1998). Additionally, the presence of particulate and dissolved organics in natural seawater may have further contributed to the inhibition of mineral precipitation (Moras et al., 2024; Naviaux et al., 2019; Subhas et al., 2018). Our study does not provide quantitative constraints on the tip because the sampling resolution was not high enough to determine the occurrence and duration of the induction period. Further, there is little data in the literature on this important feature of mineral precipitation. Future work should focus on directly constraining t<sub>ip</sub> under a wide range of OAE-relevant chemical conditions.

### 4.6 Runaway precipitation

Alkalinity addition can raise  $\Omega$  high enough to initiate spontaneous precipitation of minerals whose surfaces can then serve as additional nucleation sites for crystal growth, leading to a phenomenon referred to as "runaway" precipitation, where more alkalinity is removed than had been added (Fuhr et al., 2022; Hartmann et al., 2023; Moras et al., 2022; Schulz et al., 2023; Suitner et al., 2024). Our data suggest the occurrence of runaway precipitation in all experiments (Fig. 6A and 10B).

Examining the relationship between alkalinity addition and alkalinity decline (i.e., the decrease in TA over the course of the experiment) provides further insights into the nature of this

phenomenon (Fig. 10B). When plotting alkalinity addition on the x-axis and ΔTA on the y-axis, the horizontal line (slope and intercept = 0) indicates no precipitation whereas the 1:1 line (slope = 0) indicates that the amount of alkalinity removed due to precipitation equals the amount added. The ideal scenario is that datapoints fall on the horizontal line or at least below the 1:1 line. The experiments of Moras et al. (2022), where alkalinity was enhanced by 250 μmol.kg<sup>-1</sup> are an example of such a scenario where no alkalinity was removed due to precipitation (Fig. 10B). In contrast, all other experiments fall above the 1:1 line, reflecting runaway precipitation where more alkalinity is removed than added. The data also suggests that the intensity of the runaway precipitation increases with higher alkalinity addition (Fig. 10B).

The extrapolated  $\Omega_{\infty}$  value in our experiments is always above 1, suggesting that mineral precipitation did not lower  $\Omega$  until equilibrium is reached but ceased at an  $\Omega_A$  of  $\sim$  3 regardless of the initial  $\Omega_A$  which ranged between 10 and 30 (Fig. 10C). One explanation to why equilibrium is not reached is that precipitation becomes kinetically limited at lower  $\Omega$  values, due to the presence of Mg, phosphate, and organics, all of which are known to inhibit carbonate mineral precipitation (e.g., Berner, 1975; Mills et al., 2022). Alternatively,  $\Omega$  may not have reached equilibrium because the duration of the experiments was too short. Indeed, the experiments of Moras et al. (2022), which were conducted for longer durations than ours, are all characterized by lower final  $\Omega$  values, although two out of the three experiments analyzed here have an  $\Omega$  of  $\sim$  2, suggesting that even with long durations equilibrium may not be reached due to kinetic inhibition of mineral precipitation (Fig. 10C).

### **5 Conclusions**

This study used shipboard experiments to investigate mineral precipitation from seawater following alkalinity addition to test the efficiency of OAE as an ocean-based CO<sub>2</sub> sequestration approach. Thermodynamic calculations indicate that OAE causes seawater to become supersaturated with respect to numerous minerals. The mineralogical data reveal that the dominant mineralogy of precipitate in all experiments is aragonite followed by a minor amount of calcite. Small amount of brucite is detected only in the experiment where 2000 µmol.kg<sup>-1</sup> of alkalinity was added. Moreover, the data show that the crystallographic characteristics (crystallite size and microstrain) of aragonite evolve through time, consistent with the occurrence of Ostwald ripening.

The logarithm of the precipitation rate correlates with the logarithm of  $\Omega-1$ , yielding a reaction order of  $2.16\pm0.5$  and a rate constant of  $0.15\pm0.009~\mu mol.hr^{-1}$  for our experiments and those of Moras et al. (2022). Our reaction order is generally comparable to that derived from previous studies, but the rate constant is an order of magnitude lower. This difference is attributed to the fact that our experiments and most of those of Moras et al. (2022) were unseeded whereas previous studies (e.g., Burton and Walter, 1987; Mucci et al., 1989) used carbonate seeds that act as nuclei for precipitation. The onset of precipitation was detected after an induction period, the length of which is inversely correlated with the initial  $\Omega$ . Mineral precipitation occurred in all experiments suggesting that the threshold  $\Omega_A$  for precipitation is < 11. Precipitation took place in a runaway manner, decreasing TA to values below that of seawater prior to alkalinity addition.

Our results demonstrate that the highest risk of mineral precipitation is immediately following alkalinity addition and before dilution and  $CO_2$  uptake by seawater, both of which lower  $\Omega$ . The observation that the dominant mineralogy of precipitate is aragonite indicates that mineral precipitation has a negative impact on OAE efficiency because aragonite is unlikely to redissolve given that the surface ocean is currently supersaturated with respect to this mineral. Magnesium hydroxide (i.e., brucite) precipitation, in contrast, is less problematic because it is more likely to redissolve, releasing alkalinity back into seawater. The occurrence of runaway precipitation also means that mineral precipitation following OAE may not only decrease OAE efficiency at sequestering  $CO_2$  but can render this approach counterproductive. As such, mineral precipitation should be avoided by keeping  $\Omega$  below the threshold of precipitation and quantifying its consequences on OAE efficiency if it occurs. Lastly, in order to be able to quantitatively determine the impact of mineral precipitation on OAE under a wide range of conditions, a mechanistic understanding of precipitation in the context of OAE must be developed.

#### Code/Data availability

All data used in this study are archived in the BCO-DMO repository. This includes the carbonate chemistry measurements and precipitate kinetics data (Hashim et al., 2025a), the raw XRD data and the associated Rietveld refinement results (Hashim et al., 2025b), and the Thermogravimetric analysis (Hashim et al. 2025c).

# Acknowledgements

| 623 | We would like to thank the captains and crew members of expeditions AE2316 and                  |
|-----|-------------------------------------------------------------------------------------------------|
| 624 | AE2321. Thank you to Ben Van Mooy, the chief scientist of expedition AE2321, Helen Fredricks    |
| 625 | for helping with field work, Donna Dimarchopoulou and Charles Settens for assisting with sample |
| 626 | preparation for XRD analysis, and Dave Bish for insightful discussions regarding XRD data. MSH  |
| 627 | was funded by NSF OCE Postdoctoral Fellowship (award #2205984); AVS, LM, MH, and DM             |
| 628 | were funded by the Carbon to Sea Initiative; CD was funded by a NOAA-NOPP award (NOAA-          |
| 629 | OAR-OAP-2023-2007714); and EB was funded by NSF OCE (award #2123055). Thank you to              |
| 630 | Devon Cole and Charly Moras for providing constructive feedback that improved the study and     |
| 631 | the Associate Editor Olivier Sulpis for handling the manuscript.                                |
|     |                                                                                                 |

633

# **Competing interests**

The authors declare that they have no conflict of interest.

- 636 Author contributions
- 637 Conceptualization: MSH and AVS
- 638 Funding Acquisition: AVS, MSH, FK
- Data collection: MSH, EB, MH, FK
- 640 Investigation: MSH, AVS, LM, FK DM, CD
- Writing original draft: MSH
- Writing review and editing: MSH, AVS, DM, FK, LM, CD, MH, EB

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
