# Peer review of "Mineral Formation during Shipboard Ocean Alkalinity Enhancement Experiments in the North Atlantic"

_EGUsphere, 2025_

## Referee Comment (RC1)

Reviewer comments – BG2025988 – Mineral Formation during Shipboard Ocean Alkalinity Enhancement Experiments in the North Atlantic

Overview:

The manuscript from Hashim et al. presents results from an OAE experiment conducted with liquid alkalinity (i.e., NaOH solution) during a research expedition from 2023. Using natural seawater, the alkalinity was increased using a 1M NaOH solution and the carbonate chemistry was measured throughout the experiment. Finally, the data explore the formation of $CaCO_3$ and compare the results with available data from the literature.

Overall, the manuscript is very reader friendly. The setting, experimental design, results and discussion are easily understandable, and the data presented here match other data available from the literature. One very interesting aspect is that in the manuscript introduces for the first time (as far as the reviewer knows) the proposed carbonate chemistry sampling techniques from Schulz et al., 2023, available in the Guide to Best Practices in Ocean Alkalinity Enhancement Research. Furthermore, the authors focused on the mineralogy of precipitated material with continuous XRD analysis of the precipitated $CaCO_3$. Finally, the data were fitted with other work available in the literature (Burton and Walter, 1987, Moras et al., 2022, and Mucci et al., 1989) which allow for an easy and effective comparison of the precipitation of $CaCO_3$ under various seawater conditions. The reviewer is supportive of the publication of the research after discussing and/or addressing the various comments and questions below, and believes that after these minor revisions, the manuscript should be considered for publication.

Comments:

Lines 37-38: I agree with the terminology "unseeded". However, I am questioning the term (pseudo)homogeneously. The use of unfiltered seawater (line 25) suggests that there may be some particles in suspension, which could have acted as seed. I think that the terminology should be slightly reviewed, emphasizing in the text that the term "unseeded" refers to the absence of $CaCO_3$ seeds but that there might still be some resuspended particles that could have been used as precipitation nuclei for $CaCO_3$

Line 38: I believe the right spelling would be "homogeneous" rather than "homogenous"

Line 40: I believe the word "correlated" was intended rather than "correlate"

Line 85: for consistency, the TA unit should be reported as $\mu mol\ kg^{-1}$ throughout the text

Line 89: same as line 38

Line 105: I rather use the term magnesium hydroxide here. While I agree that brucite is the mineral form of magnesium hydroxide, $Mg(OH)_2$ can in some instances precipitate I an amorphous form which is not considered brucite. For ease, I would stick to magnesium hydroxide throughout the text

Line 115: I believe the sentence should read "… the ones that are more likely…"

Line 132: were the incubated water in the bags exposed to any movement (floating around, boat rocking, etc.) or was it considered static? Such absence or presence of movement may have affected the $CaCO_3$ precipitation kinetics and should be mentioned explicitly

Line 133: was the unfiltered seawater passed through a 1 or 2 mm mesh to get rid of bigger particles or was it fully unfiltered?

Line 147: I believe the standard notation for TA concentration is $\mu mol\ kg^{-1}$ without the "." in between. May need to be edited throughout the text

Line 154: same as line 85; also, it would be beneficial to have a column with the measured $\Delta TA$ to show the maximum TA reached, as well as indicate whether there are some discrepancies (maybe from early $CaCO_3$ precipitation after addition?)

Line 156: how was salinity measured? Because salinity does not have unit if measured on the practical salinity scale of 1978

Line 178-179: how exactly were the DIC samples taken? For stable DIC sampling, it is advised to sample the DIC in a borosilicate vial as described here using a peristaltic pump with the tubing placed at the bottom of the vial, and allowing at least half of the vial volume of overflow (Dickson et al., 2007). This section might need slightly more details.

Line 191: was the titrant ionic strength adjusted to match the samples' ionic strength?

Line 217: wouldn't calculating the various $\Omega$ at 27 °C (line 132) instead of 25 °C more suitable considering the experiments were run at ~27 °C? Or are the differences negligible?

Line 322: in caption, 3rd line, I believe there is a letter "r" missing, it should read "DIC decrease"

Line 336-337: the sentence reads that magnesite is both highly ($\Omega > 100$) and moderately ($\Omega \sim 10$) supersaturated. Please edit

Line 354: here, the figure 3 is discussed. There is one pattern that I noticed and seems interesting to me. It appears that from figure 3, the aragonite A111 and A021 signals decrease at 8.8h and 15.8h after TA addition. While I may not be the more familiar with XRD analyses, I would like to have some more details as to why there is such pattern? If all the XRD samples have been handled the same way, why is there a slight decrease at these points in time? Was $CaCO_3$ precipitation halted during these times? Or is it only a sample artefact? I am not sure whether it is worth mentioning in the manuscript, but I would like to have the authors point of view on such pattern.

Line 372: see line 105 comments. It would be more justified to use the term magnesium hydroxide here as well

Line 388-389: if $CaCO_3$ coated the $Mg(OH)_2$ crystals, could this also explain why XRD did not reveal any? Even if the XRD covers the $2\theta$ range of $Mg(OH)_2$, if these are coated with $CaCO_3$, the analysis result would only show $CaCO_3$, right?

Line 402-410: this comment does not need to be addressed, but I wanted to highlight that I really appreciated the review of the methodology and the suggested work around this unexpected loss of DIC

Line 426: same as line 38

Line 426-427: here, the work of Marion et al., 2009 could be used to determine a more accurate threshold for homogeneous precipitation given the experiment salinity and temperature

Line 433-434: same as line 38

Line 453: same as line 38

Line 463: same as line 38

Line 524-526: some works are available in the literature where they report on the inhibitory effect of various compounds under both natural and OAE setting, and could be considered as references in the manuscript (Chave and Suess, 1970, Moras et al., 2024, Pan et al., 2021, Pytkowicz, 1965)

Line 571: same as line 38

---

## Author Comment (AC1)

**Reviewer 1 (Charly Moras)**

*Overview:*

*The manuscript from Hashim et al. presents results from an OAE experiment conducted with liquid alkalinity (i.e., NaOH solution) during a research expedition from 2023. Using natural seawater, the alkalinity was increased using a 1M NaOH solution and the carbonate chemistry was measured throughout the experiment. Finally, the data explore the formation of CaCO3 and compare the results with available data from the literature.*

*Overall, the manuscript is very reader friendly. The setting, experimental design, results and discussion are easily understandable, and the data presented here match other data available from the literature. One very interesting aspect is that in the manuscript introduces for the first time (as far as the reviewer knows) the proposed carbonate chemistry sampling techniques from Schulz et al., 2023, available in the Guide to Best Practices in Ocean Alkalinity Enhancement Research. Furthermore, the authors focused on the mineralogy of precipitated material with continuous XRD analysis of the precipitated CaCO3. Finally, the data were fitted with other work available in the literature (Burton and Walter, 1987, Moras et al., 2022, and Mucci et al., 1989) which allow for an easy and effective comparison of the precipitation of CaCO3 under various seawater conditions. The reviewer is supportive of the publication of the research after discussing and/or addressing the various comments and questions below, and believes that after these minor revisions, the manuscript should be considered for publication.*

We thank the reviewer for his kind words, and we appreciate the useful and insightful feedback.

*Comments:*

*Lines 37-38: I agree with the terminology "unseeded". However, I am questioning the term (pseudo)homogeneously. The use of unfiltered seawater (line 25) suggests that there may be some particles in suspension, which could have acted as seed. I think that the terminology should be slightly reviewed, emphasizing in the text that the term "unseeded" refers to the absence of CaCO3 seeds but that there might still be some resuspended particles that could have been used as precipitation nuclei for CaCO3.*

We generally agree with the reviewer and we will revise the sentence accordingly.

*Line 38: I believe the right spelling would be "homogeneous" rather than "homogenous"*

Noted.

*Line 40: I believe the word "correlated" was intended rather than "correlate"*

Thank you. We agree and will make the change.

*Line 85: for consistency, the TA unit should be reported as μmol kg-1 throughout the text*

We will make sure the units are consistent throughout the manuscript.

*Line 89: same as line 38*

Noted.

*Line 105: I rather use the term magnesium hydroxide here. While I agree that brucite is the mineral form of magnesium hydroxide, Mg(OH)2 can in some instances precipitate I an amorphous form which is not considered brucite. For ease, I would stick to magnesium hydroxide throughout the text*

Noted. We will make sure to use the term "magnesium hydroxide" where appropriate. There are places in our manuscript where we specifically discuss minerals, their solubilities, saturation states, etc. In this case, we believe that the use of "brucite" is more appropriate but in other instances where we discuss precipitation in practical sense, we will make sure we use the term "magnesium hydroxide".

*Line 115: I believe the sentence should read "… the ones that are more likely…"*

We agree and we will change the sentence accordingly.

*Line 132: were the incubated water in the bags exposed to any movement (floating around, boat rocking, etc.) or was it considered static? Such absence or presence of movement may have affected the CaCO3 precipitation kinetics and should be mentioned explicitly*

Good point. We will add a sentence to explicitly state that the bags were floating and were allowed to move freely in the incubator.

*Line 133: was the unfiltered seawater passed through a 1 or 2 mm mesh to get rid of bigger particles or was it fully unfiltered?*

It was not filtered at all but given that we obtained the water from an open ocean setting, no sediment or large critters were present.

*Line 147: I believe the standard notation for TA concentration is μmol kg-1 without the "." in between. May need to be edited throughout the text*

Noted. We will make sure to update the units throughout.

*Line 154: same as line 85; also, it would be beneficial to have a column with the measured ΔTA to show the maximum TA reached, as well as indicate whether there are some discrepancies (maybe from early CaCO3 precipitation after addition?)*

We unfortunately didn't measure the alkalinity immediately after addition. Accordingly, the first TA and DIC datapoints in each experiment are assumed rather than measured. We will explicitly explain this in the Method section.

*Line 156: how was salinity measured? Because salinity does not have unit if measured on the practical salinity scale of 1978*

The salinity was calculated from conductivity which was measured by the CTD. We will make sure to elaborate on how salinity was calculated.

*Line 178-179: how exactly were the DIC samples taken? For stable DIC sampling, it is advised to sample the DIC in a borosilicate vial as described here using a peristaltic pump with the tubing placed at the bottom of the vial, and allowing at least half of the vial volume of overflow (Dickson et al., 2007). This section might need slightly more details.*

The sampling for DIC was done using a plastic syringe and the sample was directly injected steadily into the borosilicate vial followed by poisoning with mercuric chloride. The whole procedure took less than 15 seconds and was done in a way that minimizes gas exchange. The use of peristaltic pump and overflowing were not possible as we wanted to avoid changing the volume of the experiments too much. We will make sure to add more details to this section.

*Line 191: was the titrant ionic strength adjusted to match the samples' ionic strength?*

The titrant ionic strength was adjusted to match that of seawater but we did not consider the change in ionic strength as a result of alkalinity addition given that the change is small and variable due to precipitation.

*Line 217: wouldn't calculating the various $\Omega$ at 27 °C (line 132) instead of 25 °C more suitable considering the experiments were run at ~27 °C? Or are the differences negligible?*

We believe that the difference is minor. Nonetheless, we will make sure to recalculate $\Omega$ using a temperature of 27 °C

*Line 322: in caption, 3rd line, I believe there is a letter "r" missing, it should read "DIC decrease"*

Thank you for pointing this out. We will correct this.

*Line 336-337: the sentence reads that magnesite is both highly ($\Omega$ > 100) and moderately ($\Omega$ ~ 10) supersaturated. Please edit*

Noted. We will edit this.

*Line 354: here, the figure 3 is discussed. There is one pattern that I noticed and seems interesting to me. It appears that from figure 3, the aragonite A111 and A021 signals decrease at 8.8h and15.8h after TA addition. While I may not be the more familiar with XRD analyses, I would like to have some more details as to why there is such pattern? If all the XRD samples have been handled the same way, why is there a slight decrease at these points in time? Was CaCO3 precipitation halted during these times? Or is it only a sample artefact? I am not sure whether it is worth mentioning in the manuscript, but I would like to have the authors point of view on such pattern.*

Thank you for pointing this out and giving us the opportunity to elaborate on this. We believe that this is related to the higher amount of halite present in these two samples that makes the aragonite peaks smaller since the peak intensities reflect the relative percentage of minerals. Based on the Rietveld refinement results, amount of halite in the sample at 8.8h is 19% and the sample at 15.8h is 31% (see the associated Research data). These values are abnormally high and are likely related to residual of halite precipitated during filter drying. Rinsing the filters thoroughly with DI was challenging because the precipitate blocked the filters in most cases.

*Line 372: see line 105 comments. It would be more justified to use the term magnesium hydroxide here as well*

Noted.

*Line 388-389: if CaCO3 coated the Mg(OH)2 crystals, could this also explain why XRD did not reveal any? Even if the XRD covers the 2θ range of Mg(OH)2, if these are coated with CaCO3, the analysis result would only show CaCO3, right?*

We have thought about this possibility and made some calculations that consider the linear absorption coefficient, packing density, and sample crystal size, that revealed that the sample crystal sizes are too small and the XRD beam is capable of penetrating through them even if coating by other minerals was present. However, as you mentioned in other comments, the possibility that what precipitates is an amorphous $Mg(OH)_2$ and not brucite (mineral with a structure) could mean that the XRD would not be able to detect such as a phase. We will discuss this in our manuscript.

*Line 402-410: this comment does not need to be addressed, but I wanted to highlight that I really appreciated the review of the methodology and the suggested work around this unexpected loss of DIC*

Thank you.

*Line 426: same as line 38*

Noted.

*Line 426-427: here, the work of Marion et al., 2009 could be used to determine a more accurate threshold for homogeneous precipitation given the experiment salinity and temperature*

Thank you for providing this reference. We will look into this and incorporate it into the manuscript as appropriate.

*Line 433-434: same as line 38*

Noted.

*Line 453: same as line 38*

Noted.

*Line 463: same as line 38*

Noted.

*Line 524-526: some works are available in the literature where they report on the inhibitory effect of various compounds under both natural and OAE setting, and could be considered as references in the manuscript (Chave and Suess, 1970, Moras et al., 2024, Pan et al., 2021, Pytkowicz, 1965)*

Thank you for sharing these references. We will incorporate these references into the manuscript.

*Line 571: same as line 38.*

Noted.

---

## Author Comment (AC2)

**Reviewer 2 (Devon Cole)**

*Hashim et al. investigate the rate of mineral formation during the addition of liquid alkalinity to unfiltered seawater in a shipboard experiment to gain insight on the potential outcomes of certain methods of OAE. This paper provides a new aragonite precipitation rate for 'unseeded' scenarios, although once precipitates are present they should act as seed for further precipitation. This manuscript is clearly written, well organized, and was a pleasure to read. With minor revisions I look forward to the publication of this work.*

Thank you!

*General Comments:*

*Very curious if the unfiltered seawater could have had suspended particles of any sort that might act as seeds in this experiment. From the omega threshold/induction period time, would it be possible to compare to other experiments of homogeneous precipitation and make a guess about potential impact of such particles? Although from a real-world standpoint, regardless of mechanism, this does make the most realistic approximation of the conditions that would be faced during an actual deployment of liquid alkalinity in the ocean.*

We do believe that the particles present in our unfiltered seawater make a difference in the threshold and induction period of precipitation. The observation that the threshold for precipitation observed in our experiments is lower than that of Hartmann et al. (2023) for the filtered seawater experiments is likely because of nucleation on particles. Since we did not conduct any experiments with filtered seawater, an insightful discuss of the effect of particles is not possible. However, Hartmann et al. (2023) conducted identical experiments using filtered and unfiltered seawater (they refer to as abiotic and biotic) and demonstrate the effect of particles on the threshold of precipitation.

*Did the authors by any chance filter the control and analyze to get a sense of what might have been suspended in the seawater to start?*

We did filter the control. Visually, there was very little material on the filter, certainly not enough for any mineralogical work. We analyzed the sample for particulate inorganic carbon and found that it has 0.13 µmol/L seawater.

*Would it be possible or useful to quantify the amount of precipitate and then use an estimate of BET surface area for the aragonite and work out rates which include surface area for the latter parts of the experiment once there is precipitate present? I am wondering if that could inform on the comparison of this rate to the Mucci rate more directly. Could this also allow a better understanding of the ease of precipitation on the inside of the bag (how much less efficient is it than particles) and confirmation of the normalization scheme used? It seems that if the answer to my very first comment is that omega values got high enough without precipitation that we are confident we generated homogenous precipitation, then I would think using the inside of the bag to normalize for surface area/with the Mucci rate is a sort of false value. That is, the initial surface area was effectively zero. So to convert to units w/ surface area, the authors would have*

*to rely on the accumulation of precipitate. This is quite a tricky problem and I very much look forward to seeing the authors thoughts on this!*

Thank you for the interesting and thought-provoking question. The initial $\Omega$ was high enough, especially in the +1000 and +2000 alkalinity addition experiments, to induce homogenous nucleation, based on previous published work. As you mention, this means that the surface area was effectively zero and increased with time as these nuclei grew. Therefore, we are not sure if there is a way to constrain the evolving surface area without direct measurements and experiments that are specifically designed to test this.

*Line-by-line:*

*Ln 40: should be "correlated"*

Noted. We will change that.

*Ln 316-321: add ref to the figure here (fig 9?)*

Yes. We will refer to Figure 9 at the end of the sentence.

*Ln 385: should be "may have prevented all of the brucite from redissolving,…"*

We will change the sentence according your suggestion.

*Ln 391: define ACC up here*

Noted.

*Ln 442: reaction order stated as 2.2 but shown as 2.16 on the figure, I think should make consistent throughout*

Noted. We will make sure to fix this. We also would like to mention that we discovered a small mistake in the code that performs the regression to calculate the reaction order and constant. It seems that the data from experiment D were mistakenly excluded which led to a small error in the values of n and k. The new value for the reaction order is 2.15 (compared to 2.16) and for the reaction constant is 0.20 (compared to 0.15). We will make sure to update the manuscript and figures accordingly.

*Ln 444: Curious about the temp differences. I think this warrants a bit of discussion – either as to why the author has decided the slight differences don't matter, or if they do, how they ought to be handled to make all of these experiments comparable.*

We assume you are referring to the fact that our experiments were conducted at 27 ºC, Mucci et al. (1989) and Burton and Walter (1987) at 25 ºC, and Moras et al. (2022) at 21 ºC. We have thought about conducing a temperature correction based on available data

regarding the effect of temperature on kinetics, we decided that the correction is not necessary and could be counterproductive for two reasons. First, the temperature different between our experiments and those of Mucci et al. (1989) is only 2 ºC yet the difference in reaction order is 2.15 – 1.48 = 0.67 and the difference in reaction constant is 45 – 0.2 = 44.8. Based on the data of Burton and Walter (1987), the reaction order increases by ~ 0.066 per 1 ºC  and the reaction constant by ~ 0.5 per 1 ºC. This suggests that differences in reaction order and constant would be within the reported uncertainties and are too small to make a difference.

Regarding the experiments of Moras et al. (2022), the difference is 4 ºC with our experiments. While recalculating their data to 25 ºC might bring them in closer alignment with our data (since they generally plot lower than ours in Figure 9 and a temperature increase leads to higher reaction order (steeper slope) and higher reaction constant (intercept), we think that a correction is not warranted because the data of Burton and Walter (1987) is based on seeded experiments where the reaction constant is much higher than in our and Moras et al. (2022) experiments, so it is unclear whether this extrapolation is justified given that it would only make a small difference. We will make sure to add a few lines in our manuscript so that the reader is aware of the temperature differences between experiments.

*Ln 445: seems like the reaction orders are rounded here too*

Noted.

*Ln 457: this surface area comparison and normalization is very interesting to me, and I appreciate the detail. Given the bag material and the Subhas reference, it seems that precip on the bag should be retarded relative to if that 0.24m2 surface were carbonate. So if you convert this rate to a umol/m2/time unit, it is perhaps the case that the reaction constant should be even lower if the bag surface area is not very efficient? See the rest of my thoughts on this above*

We agree that the bag inner walls, despite being non-ideal for nucleation, did serve as nucleation sites, which perhaps impacted the reaction kinetics to some degree.

*Fig 10B – Should the Y axis be negative? Its making the point that more TA is removed by the end of the experiment than what it was to start, so the delta TA should be a negative number, right?*

No, because it represents initial TA -  final TA (after precipitation) which is always less than the initial.

*SI Data:*

*The data copied over from Mucci et al 1989 do not appear to match that paper. I believe the data in the rate column are in fact omega-1 and I am not sure what the data in the log(omega-1) column are. I think these data should be from table III in that paper. I worked to calculate the surface area normalized data which are shown in Fig 9 of this manuscript for use in my own*

*work, and have come up with the same values that are plotted in Fig 9 working directly from the Mucci text, so I believe this is just a small copying error here.*

Thank you very much for pointing this out! There was indeed a copying mistake. We will make sure to update the table with the correct data.

*As a small gift to future close readers, the authors could add which table in Mucci these data come from to column A*

The data were taken from Table III. We will make sure to mention this in the spreadsheet.